# Melatonin Supplementation for Six Weeks Had No Effect on Arterial Stiffness and Mitochondrial DNA in Women Aged 55 Years and Older with Insomnia: A Double-Blind Randomized Controlled Study

**DOI:** 10.3390/ijerph18052561

**Published:** 2021-03-04

**Authors:** Yonghwan Kim, Hee-Taik Kang, Duk-Chul Lee

**Affiliations:** 1Department of Family Medicine, Chungbuk National University Hospital, Cheongju 28644, Korea; airsantajin@gmail.com (Y.K.); kanght0818@gmail.com (H.-T.K.); 2Department of Medicine, Graduate School, Yonsei University College of Medicine, Seoul 03722, Korea; 3Department of Family Medicine, Chungbuk National University College of Medicine, Cheongju 28644, Korea; 4Department of Family Medicine, Severance Hospital, Yonsei University College of Medicine, Seoul 03722, Korea

**Keywords:** melatonin, arterial stiffness, mitochondrial DNA, insomnia

## Abstract

Melatonin is a hormone produced in the pineal gland that controls sleep and circadian rhythm. Some studies have reported antioxidant and anti-inflammatory effects of melatonin that could benefit cardiometabolic function; however, there is a lack of evidence to support these assertions. The aim of this study was to investigate whether melatonin has beneficial effects on arterial stiffness and mitochondrial deoxyribonucleic acid (DNA) in humans. Methods: This study was designed as a double-blind randomized controlled study. Thirty-eight healthy women aged 55 years and older were enrolled. All had insomnia (Pittsburgh Sleep Quality Index (PSQI) ≥ 5), not treated with any medications, for at least three months before enrollment. Subjects were divided into a melatonin and a placebo group according to melatonin supplementation. The melatonin group took 2 mg melatonin every night for six weeks. The cardio–ankle vascular index (CAVI) was used as an indicator of arterial stiffness. After six weeks, CAVI, mitochondrial DNA (mtDNA) copy number in white blood cells (WBCs), and other metabolic indices, such as homeostasis model assessment of insulin resistance (HOMA-IR), were checked. Results: Sleep quality index using PSQI was improved in the melatonin group from a score of 11 to 8 (*p* = 0.01), but did not change significantly in the control group. However, there was no significant intergroup difference in PSQI. Systolic blood pressure (SBP) decreased in the melatonin group from 135 to 128 mmHg (*p* = 0.015), while remaining stable in the placebo group. Right CAVI, mitochondrial DNA copy number, and HOMA-IR were not altered in either group. There were no intergroup differences in CAVI, mtDNA, HOMA-IR, or SBP between baseline and week six. Conclusions: We found no evidence that melatonin supplementation improved cardiometabolic parameters like arterial stiffness, mtDNA, or insulin resistance compared to the placebo between baseline and week six. Sleep quality was improved in the melatonin group. Further research, including longer-term studies with higher doses of melatonin, is warranted.

## 1. Introduction

Melatonin (*N*-acetyl-5-methoxytryptamine) is a neuropeptide hormone derived from the pineal gland. It conveys signals to distant organs, principally the brain. The main function of melatonin is to control sleep and circadian rhythm, leading to improved sleep quality [1]. The potential usefulness of melatonin in cardiometabolic dysfunction is a popular topic of research, but in vivo evidence for melatonin benefits beyond sleep improvement remains scarce. Melatonin can play a role in scavenging free radicals and oxidative stress [2]. The indoleamine ring of melatonin and its metabolite detoxify free radicals [3,4]. Biosynthesis of glutathione, a potent antioxidant, is induced by melatonin [4,5]. Melatonin may also play a role in modulating the immune system and cell aging [4,6].

According to a systematic review and meta-analysis by Mohammadi-Sartang et al., melatonin supplementation significantly reduces triglycerides and total cholesterol levels, which were more evident at higher doses and longer duration [7]. Some studies reported that melatonin supplementation could reduce cardiovascular–metabolic diseases, such as hypertension, diabetes mellitus, and obesity [8,9,10,11]. Korkmaz et al. proposed that antioxidant and anti-inflammatory effects, as well as regulation of the autonomic nervous system (ANS), might play a role in preventing cardiovascular–metabolic diseases [12]. Acuna-Castroviejo et al. and Leon et al. suggested that stabilization of the mitochondrial function influences arteriosclerosis, a modifiable risk factor for cardiovascular diseases, by regulating chronic, low-grade inflammation, oxidative stress, and vascular endothelial dysfunction [13,14].

We hypothesized that melatonin supplementation could improve cardiometabolic indices in addition to sleep disturbance. Thus, we aimed to investigate the effects of melatonin supplementation on arterial stiffness and cardiovascular risk factors in women aged 55 years and older with insomnia. We also examined quantitative changes in mitochondrial deoxyribonucleic acid (DNA) (mtDNA) copy number and the function of ANS before and after melatonin supplementation. 

## 2. Materials and Methods

### 2.1. Subject

This study was a single-center, double-blind, placebo-controlled, randomized clinical trial. All subjects were women over 55 years old with insomnia (Pittsburgh Sleep Quality Index (PSQI) ≥ 5) [15] who had not taken medication for depression, insomnia, or tranquilization in the past three months. 

Exclusion criteria were as follows: a history of menopausal hormone replacement therapy; cerebrovascular diseases (including ischemic stroke and cerebral hemorrhage), cardiovascular diseases (including unstable angina, myocardial infarction, and coronary revascularization); chronic liver disease (including chronic hepatitis, liver cirrhosis, and hepatocellular carcinoma); chronic renal disease (including previous chronic kidney disease and kidney transplantation); malignant neoplasm; any treatment for depression, insomnia, or tranquilization at least three months before this study; aspartate aminotransferase (AST) > 100 IU/L, alanine aminotransferase (ALT) > 100 IU/L, or creatinine > 1.4 mg/dL.

After the initial screening visit (week zero), subjects were randomly allocated to two groups: the melatonin and placebo groups. The melatonin group received 2 mg of prolonged-release melatonin (Circadin 2 mg PR^®^) (Kuhnil Pharmacy, Seoul, Korea). The placebo group received a placebo drug with the same appearance as the melatonin pill. All subjects were instructed to take the pill once a day, two hours before sleep. The study duration was set to six weeks. After six weeks, a physical examination, questionnaire, laboratory tests, and arteriosclerosis and ANS function testing were performed. By this sixth week, subjects brought any remaining medicine and the researcher evaluated the compliance and side effects of the drug.

The Institutional Review Board at Severance Hospital, Yonsei University College of Medicine, approved this study (YUHS 4-2016-0418). The study population was recruited via advertisements posted by the Department of Family Medicine at Severance Hospital, from October 2017 to October 2018. All subjects provided written informed consent.

### 2.2. Measurement

We also collected personal lifestyle information (age, marital status, smoking, alcohol, exercise, and 15-geriatric depression scale), anthropometric measurements, blood for laboratory testing (white blood cells (WBC), red blood cells (RBC), hemoglobin, hematocrit, platelets, AST, ALT, gamma-glutamyl transferase (GGT), creatinine, total cholesterol, triglyceride, high-density lipoprotein (HDL), low-density lipoprotein (LDL), glucose, insulin, c-reactive protein (CRP), mitochondrial DNA (mtDNA)), and PSQI at the initial (week zero) and final (week six) visits. Blood samples were collected after an eight-hour overnight fasting period. WBC, RBC, hemoglobin, hematocrit, and platelets were measured using the ADVIA 2120i hematology system (SIMENS Healthcare, Diagnostics, Deerfield, IL, USA). AST, ALT, GGT, creatinine, total cholesterol, triglyceride, HDL, LDL, glucose, insulin, and CRP were measured with a Hitachi 7600 analyzer (Hitachi High Technologies Co., Tokyo, Japan).

We divided alcohol consumption into three categories: “light,” “social,” and “heavy.” Light drinkers were defined as drinking 1–2 units on any single day or one to two times a week, social drinkers as drinking 3–4 units on any single day or three times a week, and heavy drinkers as drinking more than 5 units on any single day or more than five times a week. All subjects were never-smokers. Exercise was defined as “none,” “sometimes,” or “often.” “None” was defined as no physical exercise, “sometimes” as physical exercise performed less than three times a week for 30 min or less than 150 min per week, and “often” as physical exercise performed more than three times a week for 30 min or more than 150 min per week.

Height (cm) and weight (kg) were measured, with subjects wearing light indoor clothing without shoes, using an automated stadiometer (DS-102, DONG SAHN JENIX Co., Seoul, Korea). Body mass index (BMI, kg/m^2^) was obtained as weight (kg) divided by height (m) squared. Waist circumference (cm) was checked at the umbilicus level when subjects were standing, using an elastic tape measure. Blood pressure (mmHg) was measured from the right arm, with the subject in a sitting position after he or she had rested for at least 10 min using an automatic blood pressure monitor (EASY X 800R^®^, Jawon Medical, Gyeongsan, Korea). 

We estimated insulin resistance using the homeostasis model assessment of insulin resistance (HOMA-IR) [16]. The formula of HOMA-IR is

HOMA-IR = [{fasting insulin (mU/mL) × fasting glucose (mg/dL)}/405]



#### 2.2.1. Measurement of Arterial Stiffness

Arterial stiffness was measured with a VaSera VS-1000 instrument (Fukuda Denshi Co. Ltd., Tokyo, Japan) using the cardio–ankle vascular index (CAVI) instrument [17]. Subjects were examined in the supine position after 5 min of bed rest. Electrocardiogram electrodes were placed on both wrists and a phonocardiogram was placed at the right sternum border of the second intercostal space. Cuffs were applied to both upper arms and ankles to detect brachial and ankle pulse waves. Pulse wave velocity was measured by dividing the pulse wavelength by the time needed for the pulse wave to propagate from the aorta, through the femoral artery, to the tibial artery of the ankle. CAVI was calculated as follows

CAVI = a {(2ρ/∆P) × ln (Ps/Pd) PWV^2^} + b
where Ps = systolic pressure, Pd = d iastolic pressure, ∆P = Ps − Pd, PWV = pulse wave velocity, ρ = blood density, and a and b are constants. 

We recorded mean values for right and left CAVIs. 

#### 2.2.2. Measurement of Mitochondrial DNA Copy Number

Blood samples were drawn from the peripheral vein of each participant the morning after overnight fasting, and were collected in tubes containing ethylenediaminetetraacetic acid (EDTA)(1 mg/mL). Buffy coats were separated from blood samples by centrifugation for 10 min at 3000 rpm. The supernatant was carefully discarded by pipetting. DNA was extracted using a Gentra DNA Extraction Kit (Qiagen, Hilden, Germany). Fluorescence-based quantitative polymerase chain reaction (qPCR) was used to determine mtDNA copy number in human leukocytes, according to the nuclear DNA method developed by Wong et al. [18], with some modifications.

For the determination of nuclear DNA, the forward primer 5′-GGCTCTGTGAGGGATATAAAGACA-3′ and reverse primer 5′-CAAACC-ACCCGAGCAACTAATCT-3′ (complementary to the sequences of the chromosome 1 (Chr1) genome loci on 1q24-25) were used to amplify a 97-bp product. For analysis of mtDNA, we used NADH dehydrogenase subunit 2 (*ND2*) gene sequences. The forward primer 5′-CACAGAAGCTGCCATCAAGTA-3′ and reverse primer 5′-CCGGAGAGTATATTGTTGAAGAG-3′ amplified an 89-bp product. The quantitative real-time PCR was performed in a Roche LightCycler 480r (Roche Applied Science, Mannheim, Germany) apparatus using the LightCycler 480 SYBR Green I Master kit (Roche Applied Science, Mannheim, Germany). DNA (10 ng) was mixed with 10 μL LightCycler 480 SYBR Green I Master Mix that contained 5 μmol (final concentration 0.4 μM) of forward and reverse primer, with a final volume of 20 μL. PCR reactions were conducted as follows: initiation at 50 °C for 2 min, 95 °C for 1 min, 40 cycles of denaturation at 95 °C for 15 s, annealing at 60 °C for 20 s, extension at 72 °C for 15 s, and finally holding at 25 °C. The threshold cycle number (Ct) of the *Chr1* gene and the *ND2* gene was determined for each individual qPCR run. ΔCt [Ct (*ND2*) − Ct (*Chr1*)] represents relative abundance. The quantitative results were expressed as the copy number of mtDNA/cell by 2 × 2^−ΔCt^. Each measurement was carried out at least three times and normalized in each experiment against serial dilutions of a control DNA sample [18,19].

#### 2.2.3. Statistical Analysis

The intended sample of 40 recruited subjects (20 melatonin and 20 placebo) provided a power of approximately 80%, assuming a significance level of <0.05 between the melatonin and placebo groups. The dropout rate was expected to be less than 10% based on the good tolerability of pharmaceuticals. Ultimately, 38 subjects were enrolled. We used nonparametric statistical analysis because of enrollment numbers. Data are presented as median (interquartile range (IQR)) or number (percentage). Wilcoxon signed-rank test was used to compare intragroup differences before and after intervention.

The Mann–Whitney U test was used to calculate the difference between two groups before and after intervention. We conducted all analyses using SPSS version 23.0 (SPSS Inc., Chicago, IL, USA). All statistical tests were two-sided, and a value of *p* < 0.05 was considered statistically significant.

## 3. Results

Figure 1 demonstrates the schema of this study and the criteria for inclusion and exclusion. Fifteen subjects were excluded. Three applicants did not meet the PSQI score. Six applicants with a history of malignant neoplasms, four applicants on medications for depression within three months prior to the study’s initiation, and two applicants who withdrew informed consent due to familial opposition to this study were excluded. Ultimately, 38 subjects were included in final analyses. After exclusion, all enrolled subjects fully participated in this study without the discontinuation of medication. One woman had a side effect of abdominal discomfort that was resolved without intervention. The medication possession rate was over 80% for all subjects. 

Baseline clinical characteristics of 38 subjects are shown in Table 1. All parameters were similar between the placebo and melatonin groups, except total cholesterol and insulin levels at baseline. Total cholesterol was higher in the melatonin group than in the placebo group (*p* = 0.031), while insulin was higher in the placebo group than in the melatonin supplement group (*p* = 0.030). 

Table 2 shows changes in cardiovascular parameters, sleep index, insulin resistance, and mtDNA. We compared intra- and inter-group differences in CAVI, mtDNA, HOMA-IR, PSQI, and SBP between the baseline and week six in the placebo and melatonin groups. Sleep quality index improved in the melatonin group (*p* = 0.010), but did not change significantly in the control group (*p* = 0.230). However, there were no significant intergroup differences between the melatonin and placebo groups (*p* = 0.158). SBP was reduced from 135 to 128 mmHg in the melatonin group (*p* = 0.015), while SBP in the placebo group was not altered (*p* = 0.153). Right CAVI, mtDNA copy number, and HOMA-IR were not affected in either group. Left CAVI was elevated in the placebo group. There were no intergroup differences in CAVI, mtDNA, HOMA-IR, or SBP from the baseline to week six between the placebo and melatonin groups.

Table 3 shows other metabolic indices. Intra- and inter-group differences in geriatric depression scale, pulse rate, BMI, WBC, hemoglobin, ALT, GGT, creatinine, total cholesterol, LDL cholesterol, glucose, CRP, and heart rate variability indices were demonstrated. Pulse rate was higher in the melatonin group (*p* = 0.021), but not significantly changed in the placebo group (*p* = 0.420). Intergroup difference was observed between the two groups at the baseline and week six (*p* = 0.019). Additionally, hemoglobin changes were not significant in intragroup comparisons but were significantly different in intergroup comparisons (*p* = 0.036). There were no intra- and inter-group differences in other parameters (all *p* > 0.05).

## 4. Discussion

This double-blind, randomized, placebo-controlled study showed that a six-week melatonin intervention improved sleep quality and decreased SBP levels in women aged 55 years and older with insomnia. However, melatonin supplementation did not improve arteriosclerotic markers, such as CAVI, or insulin resistance.

Lemoine reported that prolonged-release melatonin significantly improved quality of sleep and morning alertness in women with primary insomnia [20]. Scheer demonstrated that regular melatonin supplementation at bedtime reduced nocturnal blood pressure in men with essential hypertension [21]. According to Możdżanstudy [22], the blood-pressure-lowering effects of melatonin are statistically significant in individuals with type 2 diabetes with non-dipper type hypertension. These findings are consistent with the current study, in which melatonin improved sleep quality and reduced SBP.

Tengattini suggested that several cardiac conditions, such as cardiac ischemia/reperfusion, atherosclerosis, and hypertension, were a consequence of free radical damage and processes involving an inflammatory response [23]. The beneficial effects of melatonin administration against pro-inflammatory conditions may be directly and indirectly due to its free radical-scavenging activities and anti-oxidative function. When exogenously administered, melatonin is quickly distributed throughout the organism, where it moves into the intracellular area, and is accumulated in the mitochondria. Mitochondria are a major site of free radical and oxidative stress generation. Melatonin protects against mitochondrial dysfunction through the prevention of mitochondrial membrane depolarization by reactive oxygen species (ROS) [24]. Melatonin can serve as a beneficial agent promoting some clinical conditions related to mitochondrial function. Obayashi [25] and Lee [26] supported the hypothesis that urinary-excreted melatonin metabolite levels are inversely associated with arterial stiffness. Initially, we hypothesized that melatonin would improve CAVI as an auxiliary indicator for arterial stiffness [27,28], mitochondrial DNA, and HOMA-IR, as well as the other cardiometabolic markers beyond its effects on insomnia.

Arterial stiffness, insulin resistance, and mtDNA copy number were not improved after six weeks of supplementation with melatonin in this study. This is inconsistent with previous studies and our hypothesis. Unlike the abovementioned studies, this study was designed as a randomized, placebo-controlled, double-blinded trial in a hospital setting. It is possible that, compared to observational studies or non-randomized, controlled, clinical trials, this randomized, controlled, clinical study is more stringently designed and better reflects the pharmaceutical effects of melatonin in human subjects. 

In part, the origin of these differences may be randomization and allocation at baseline in relation to the small sample size. Six subjects (31.6%) in the placebo group and eight subjects (42.1%) in the melatonin group had dyslipidemia and took anti-dyslipidemic medications. Six subjects (31.6%) in the placebo group and one subject (5.3%) in the melatonin group had hypertension. The small sample size may have contributed to this null association between melatonin supplements and clinical outcomes. 

Our study has some limitations that should be considered when interpreting these results. Six weeks may be an insufficient duration to observe the effects of improving arterial stiffness, mtDNA, and insulin resistance. Few studies with a randomized controlled design have investigated the duration of melatonin use and cardiometabolic function. The follow-up duration of most previous studies on melatonin’s sleep improvement effects was less than four weeks [11,20,29,30,31]. In the few studies that have investigated its cardiometabolic effects, the follow-up did not exceed six weeks. Thus, we considered a follow-up duration of six weeks to be appropriate for investigating the cardiometabolic effects of melatonin. Due to the lack of evidence regarding long-term use, we set the treatment period to six weeks. It is possible that longer-term use could reveal further effects that were not observed in this study, and further research is warranted. Additionally, a prolonged-release 2 mg melatonin dose might have been too low to reverse arterial stiffness. We could not escalate the dose because of dose restrictions related to Circadin^®^ 2 mg and its approval by the Korean Ministry of Food and Drug Safety for melatonin. Future research should investigate higher melatonin doses and their effects on cardiovascular factors and mitochondrial function. In addition, urine metabolites were not measured before and after the experiment to determine the concentration of active melatonin in the human body. Circulating melatonin is inactivated in the liver where it is converted to 6-hydroxymelatonin and then conjugated with sulfate or glucuronide and excreted in urine. Circulating melatonin levels have been reported to decrease with physiological aging [32]. If the amount of active melatonin was measured, we could more accurately compare the effects of melatonin on cardiometabolic function. Finally, all subjects were generally healthy. Therefore, caution is needed when generalizing these results to people with chronic severe illness.

This study has several advantages over previous research. This study was carefully designed as a randomized, placebo-controlled, double-blinded trial that enrolled 38 women. Few randomized, controlled, double-blind trials of melatonin supplements have been conducted. Among them, most studies enrolled less than 20 participants. Population size is important because it affects statistical power. While the number of participants in this study was small, it was larger than previous studies.

## 5. Conclusions

Consistent with previous studies, we have shown that melatonin supplementation for six weeks improves sleep quality. However, it was not proven that melatonin could improve cardiometabolic parameters such as arterial stiffness, mtDNA, or insulin resistance in this human study. In part, this null association between melatonin supplementation and cardiometabolic indices might be due to the low dose or short duration of melatonin supplementation. Further investigation with a long-term intervention at a higher dose of melatonin is needed to obtain a clear understanding of the cardiometabolic effects of melatonin treatment.

## Figures and Tables

**Figure 1 ijerph-18-02561-f001:**
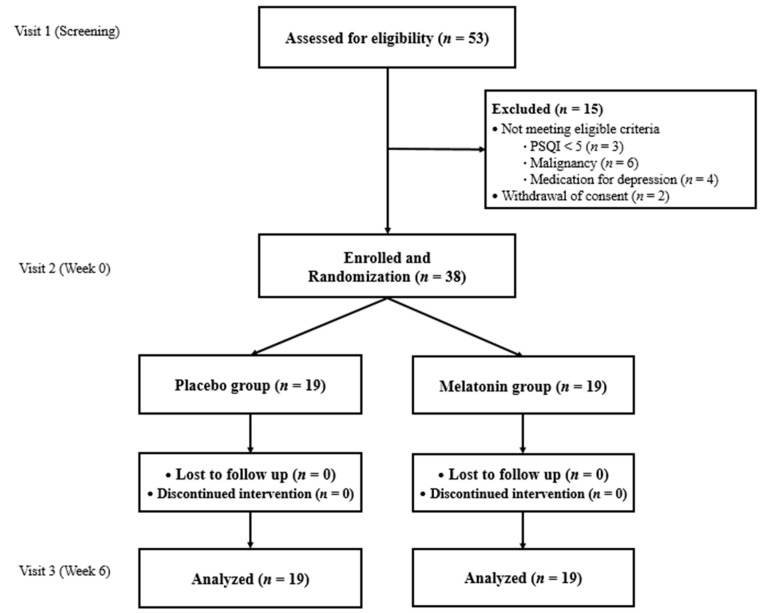
Flow diagram representing the study selection.

**Table 1 ijerph-18-02561-t001:** Baseline characteristics according to intervention.

Variables	Placebo	Melatonin	*p*-Value
Age, years	61.00 (59.00, 65.00)	61.00 (58.00, 71.00)	0.204
Alcohol consumption, N (%)	19 (100)	19 (100)	0.183
Soft	10 (52.63)	5 (26.32)	
Social	0 (0)	1 (5.26)	
Heavy	9 (47.37)	13 (68.42)	
Physical activity, N (%)	19 (100)	19 (100)	0.403
None	2 (10.53)	1 (5.26)	
Sometimes	7 (36.84)	4 (21.05)	
Often	10 (52.63)	14 (73.69)	
MPR, score	97.62 (4.76)	97.62 (4.76)	0.988
PSQI, score	12.00 (6.00)	11.00 (2.50)	0.576
GDS, score	3.00 (6.00)	1.00 (4.00)	0.114
SBP, mmHg	129.00 (28.00)	135.00 (15.00)	0.082
DBP, mmHg	77.00 (13.00)	77.00 (10.00)	0.861
Weight, kg	57.60 (8.80)	62.00 (10.60)	0.274
BMI, kg/m^2^	23.60 (4.40)	24.90 (3.20)	0.365
Waist, cm	80.00 (15.00)	82.00 (10.00)	0.988
GGT, mg/dL	17.00 (12.00)	16.00 (10.00)	0.781
Cr, mg/dL	0.62 (0.11)	0.68 (0.18)	0.220
Total cholesterol, mg/dL	178.00 (73.00)	209.00 (40.00)	0.031
TG, mg/dL	106.00 (33.00)	107.00 (67.00)	0.885
HDL cholesterol, mg/dL	51.00 (7.00)	53.00 (19.00)	0.385
LDL cholesterol, mg/dL	101.80 (52.20)	129.80 (51.60)	0.109
Glucose, mg/dL	96.00 (8.00)	98.00 (10.00)	0.885
Insulin, uU/mL HOMA-IR	6.30 (4.96) 1.68 (1.29)	4.37 (2.87) 1.04 (0.78)	0.030 0.068
CRP, mg/L	0.60 (0.70)	0.60 (0.50)	0.481

Abbreviations: MPR, medication possession rate; PSQI, Pittsburgh Sleep Quality index; GDS, 15-item Geriatric depression scale; BMI, body mass index; SBP, systolic blood pressure; DBP, diastolic blood pressure; Wt, Weight; BMI, Body Mass Index; GGT, gamma-glutamyl transferase; Cr, creatinine; TG, triglyceride; HDL, high-density lipoprotein; LDL, low-density lipoprotein; HOMA-IR, Homeostatic Model Assessment for Insulin Resistance; CRP, C-reactive protein. Alcohol consumption: Light, 1–2 units on any single day or 1–2 times/week; Social, 3–4 units on any single day or 3 times/week; Heavy, more than 5 units on any single day or more than 5 times/week. Physical activity: Sometimes, less than three times a week for 30 min or less than 150 min per week; Often, more than three times a week for 30 min or more than 150 min per week. All values are presented as median (interquartile range) or number (percentage). *p*-values were derived from the chi-square test for smoking, alcohol, and exercise. *p*-values were derived from the Mann–Whitney U test for age, MPR, PSQI, GDS, SBP, DBP, weight, BMI, waist, GGT, Cr, total cholesterol, TG, HDL, LDL, glucose, insulin, and CRP.

**Table 2 ijerph-18-02561-t002:** Changes in cardiovascular parameters, sleep index, insulin resistance, and mitochondrial DNA.

	Placebo (*n* = 19)			Melatonin (*n* = 19)			
Variable	Baseline	6 Weeks		Baseline	6 Weeks		Intergroup Difference
	Median (IQR)		*p*-Value ^a^	Median (IQR)		*p*-Value ^a^	*p*-Value ^b^
RCAVI	8.10 (1.25)	7.80 (0.12)	0.587	8.05 (2.00)	7.90 (1.83)	0.163	0.582
LCAVI	8.15 (1.78)	8.20 (1.27)	0.025	7.60 (1.28)	7.45 (1.30)	0.095	0.813
mtDNA, copy number	43.70 (20.50)	42.40 (21.60)	0.223	59.40 (50.80)	44.20 (28.00)	0.306	0.953
HOMA-IR	1.68 (1.29)	2.01 (1.15)	0.260	1.04 (0.78)	1.17 (0.84)	0.573	0.661
PSQI, score	12 (6)	9 (6)	0.230	11 (3)	8 (5)	0.010	0.158
SBP, mmHg	129 (28)	121 (15)	0.153	135 (15)	128 (16)	0.015	0.781

Abbreviations: IQR, interquartile range; RCAVI, Right cardio–ankle vascular index; LCAVI, Left cardio–ankle vascular index; mtDNA, mitochondrial DNA copy number; HOMA-IR, Homeostasis model assessment of insulin resistance; PSQI, Pittsburgh Sleep Quality index; SBP, systolic blood pressure. ^a^
*p* values were calculated by Wilcoxon signed-rank test for intragroup differences before and after intervention. ^b^
*p* values for intergroup difference were calculated by the Mann–Whitney U test for the difference between two groups before and after intervention.

**Table 3 ijerph-18-02561-t003:** Comparison of other metabolic indices.

	Placebo (*n* = 19)			Melatonin (*n* = 19)			
Variable	Baseline	6 Weeks		Baseline	6 Weeks		Intergroup Difference
	Median (IQR)		*p*-Value ^a^	Median (IQR)		*p*-Value ^a^	*p*-Value ^b^
GDS, score	3 (6)	1 (3)	0.001	1 (4)	1 (2)	0.076	0.165
PR	71 (12)	71 (11)	0.420	72 (12)	77 (8)	0.021	0.019
BMI, kg/m^2^	23.60 (4.40)	23.90 (5.00)	0.635	24.90 (3.20)	25.10 (2.70)	0.793	0.558
WBC, 10^3^/μL	6010 (930)	5780 (1540)	0.747	4980 (1440)	5690 (1350)	0.083	0.293
Hb, g/dL	13.10 (1.30)	13.10 (1.40)	0.266	13.30 (1.30)	13.40 (1.30)	0.081	0.036
ALT, IU/L	17 (9)	17 (11)	0.588	20 (8)	19 (11)	0.686	0.500
GGT, mg/dL	17 (12)	16 (12)	0.230	16 (10)	16 (9)	0.152	0.054
Cr, mg/dL	0.62 (0.11)	0.63 (0.04)	0.601	0.68 (0.18)	0.73 (0.17)	0.247	0.640
T.chol, mg/dL	178 (73)	190 (61)	0.170	209 (40)	215 (43)	0.421	0.492
LDL, mg/dL	101.8 (52.2)	113.8 (47.6)	0.546	129.8 (51.6)	139.0 (57.2)	0.702	0.872
Glucose, mg/dL	96 (8)	99 (15)	0.221	98 (10)	97 (18)	0.635	0.169
CRP, mg/L	0.60 (0.70)	0.70 (0.70)	0.241	0.60 (0.50)	0.50 (0.60)	0.275	0.140
SDNN	26.04 (11.77)	25.84 (17.78)	0.936	31.72 (22.64)	30.47 (12.99)	0.420	0.564
VLF	5.07 (0.96)	5.19 (1.20)	0.573	5.45 (1.31)	5.54 (1.19)	0.777	0.671
LF	4.40 (1.19)	4.24 (1.51)	0.334	4.86 (0.97)	4.47 (0.78)	0.557	0.153
HF	4.64 (1.75)	4.73 (1.75)	0.778	4.83 (1.61)	4.89 (0.89)	0.945	0.832
TP	6.11 (1.07)	6.09 (1.12)	0.324	6.28 (0.69)	6.25 (0.76)	0.586	0.346
LF/HF ratio	0.84 (0.78)	0.78 (2.16)	0.198	0.90 (1.17)	0.86 (1.39)	0.472	0.693

Abbreviations: IQR, interquartile range; MPR, medication possession rate; GDS, 15-item Geriatric depression scale; BMI, body mass index; DBP, diastolic blood pressure, TG, triglyceride; HDL, high-density lipoprotein; LDL, low-density lipoprotein; CRP, C-reactive protein; SDNN, standard deviation of all normal-to-normal interval; VLF, very low frequency; HF, high frequency; LF, low frequency; TP, total power. ^a^
*p* values were calculated by Wilcoxon signed-rank test for intragroup differences before and after intervention. ^b^
*p* values for intergroup difference were calculated by Mann–Whitney U test for the difference between two groups before and after intervention.

## Data Availability

Not applicable.

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
