# Peer review of "Melatonin Supplementation for Six Weeks Had No Effect on Arterial Stiffness and Mitochondrial DNA in Women Aged 55 Years and Older with Insomnia: A Double-Blind Randomized Controlled Study"

_ijerph, 2021, doi:10.3390/ijerph18052561_

Round 1

Reviewer 1 Report

The authors have now addressed my previous concerns.

This manuscript is a resubmission of an earlier submission. The following is a list of the peer review reports and author responses from that submission.

Round 1

Reviewer 1 Report

The manuscript presents an interesting study where authors conducted a double-blind randomized controlled study on insomniac women 55 years or older to understand the effect of melatonin on arterial stiffness and mitochondrial DNA. The results indicate that melatonin treatment was helpful in improving sleep quality and reducing systolic blood pressure. However, it did not have any significant effect on indicators of arterial stiffness, mitochondrial DNA copy number, copy number of WBC and other metabolic indicators. 

The manuscript is well written and clear. The study design and statistical methods are appropriate. However, the authors did not explain well why they choose melatonin intervention length of 6 weeks? So, authors should explain clearly in the introduction section why they think that 6 weeks length is appropriate to determine the effect of melatonin on arterial stiffness? 

Author Response

We agree with your opinion. Unfortunately, we found few studies on the optimal duration for melatonin use when the study was designed. According to one study using our supplement (Circadin®), patients with sleep disorders who took melatonin for three weeks exhibited improved sleep quality. (Lemoine P et al. Prolonged-release melatonin improves sleep quality and morning alertness in insomnia patients aged 55 years and older and has no withdrawal effects. J Sleep Res. 2007 Dec;16(4):372-80.) Another double-blind randomized controlled study reported that melatonin supplement for three weeks significantly improved both quality of sleep and morning alertness compared to placebo. (Wade AG et al. Efficacy of prolonged release melatonin in insomnia patients aged 55-80 years: quality of sleep and next-day alertness outcomes. Curr Med Res Opin. 2007 Oct;23(10):2597-605)

Recently, a meta-analysis of melatonin on primary sleep disorders was conducted that analyzed 19 trials. Among these studies, the follow-up duration of 16 trials was ≤28 days.

This meta-analysis reported that melatonin supplementation was beneficial in terms of sleep onset latency, total sleep time, and sleep quality. (Ferracioli-Oda E et al. melatonin for the treatment of primary sleep disorders. PLoS One. 2013 May 17;8(5):e63773)

Also, according to a meta-analysis of the effects of melatonin on nocturnal blood pressure by Ehud Grossman et al., six trials had a duration of less than four weeks duration and one trial a follow-up duration of 90 days. Four of six trials found statistically significant results.

(Grossman, Ehud et al. Effect of melatonin on nocturnal blood pressure: meta-analysis of randomized controlled trials. Vascular health and risk management vol. 7 (2011): 577-84.)

An animal study demonstrated that melatonin administration reduced media thickness of mesenteric small arteries and total aortic collagen content in rats over a period of six weeks. (Rezzani, R. et al. Effects of melatonin and Pycnogenol on small artery structure and function in spontaneously hypertensive rats. Hypertension, 2010, 55(6), 1373-1380.).

 Based on the above-mentioned studies, we expected a duration of 6 weeks to be adequate for observing melatonin-induced changes in cardiovascular risk factors such as arterial stiffness.

In general, melatonin levels drop quickly due to its short half-life (30-40 minutes), so sufficient melatonin is not maintained throughout sleep. On the other hand, Circadin®, which is similar to the endogenous melatonin release form, slowly releases melatonin for 8 to 10 hours, and works throughout the sleep period without breaking the body's sleep rhythm. For this reason, we anticipated better effects. Now that the study was done, I think we would have had better results if we had considered the accumulated effects of melatonin. When we searched for articles about effect of long-term use of melatonin supplements, the effects of long-term use have not been studied and remain unclear. A well-designed study on longer use of melatonin is needed.

We explained this at the discussion section like below:

Discussion:

Our study has some limitations that should be considered when interpreting these results. Six weeks may be an insufficient duration to observe the effects of improving arterial stiffness, mtDNA, and insulin resistance. Few studies with a randomized controlled design have investigate the duration of melatonin use and cardiometabolic function. The follow-up duration of most previous studies on melatonin’s sleep improvement effects was less than four 4 weeks.[11, 20, 29-31] In the few studies that have investigated its cardiometabolic effects, follow-up did not exceed six weeks. Thus, we considered a follow-up duration of six weeks appropriate for investigating the cardiometabolic effects of melatonin. Due to the lack of evidence on long-term use, we set the treatment period to 6 weeks. It is possible that longer-term use could reveal further effects that were not observed in this study, and further research is warranted. (e.g. line 45-49, Page 8)

Citation>

  1. Grossman, Ehud et al. Effect of melatonin on nocturnal blood pressure: meta-analysis of randomized controlled trials. Vascular health and risk management vol. 7 (2011): 577-84.
  2. Lemoine P, Nir T, Laudon M, Zisapel N. Prolonged-release melatonin improves sleep quality and morning alertness in insomnia patients aged 55 years and older and has no withdrawal effects. J Sleep Res. 2007 Dec;16(4):372-80.
  3. Wade AG, Ford I, Crawford G, McMahon AD, Nir T, Laudon M, Zisapel N. Efficacy of prolonged release melatonin in insomnia patients aged 55-80 years: quality of sleep and next-day alertness outcomes. Curr Med Res Opin. 2007 Oct;23(10):2597-605.
  4. Rezzani, R., Porteri, E., De Ciuceis, C., Bonomini, F., Rodella, L. F., Agabiti Rosei, E. Effects of melatonin and Pycnogenol on small artery structure and function in spontaneously hypertensive rats. Hypertension, 2010, 55(6), 1373-1380.
  5. Ferracioli-Oda E et al. melatonin for the treatment of primary sleep disorders. PLoS One. 2013 May 17;8(5):e63773

Reviewer 2 Report

The manuscript by Kim and co-workers elaborates on the clinical benefit of supplementation of the pineal hormone melatonin. The authors investigated 38 healthy women, 55 years and older with a recorded history of insomnia. The authors supplemented at nightime only 2mg melatonin, which leads to a serum concentration that is in the upper physiological range. The authors report a beneficial effect on the sleep architecture of participants, and a small effect on blood pressure. However, they could not show any result of melatonin supplementation on arteriosclerotic parameters, nor on mitochondrial DNA copy number. The authors conclude that melatonin has obviously no effect on cardiometabolic parameters. The study is well designed, data are valid and well presented, with minor changes to be incorporated.

- in my eyes the anti-oxidative, anti-inflammatory and anti-apoptotic effect of melatonin is largely overestimated. A strong proof of evidence is in my eyes still lacking. Therefore, citations #2-#10 should be taken with caution. I rather recommend that the authors cite solid studies that see the claimed effects of melatonin in a very critical light! While melatonin has a clear-cut effect on the phase of the endogenous circadian clock and on sleep architecture, other attributed effects are standing on week grounds. Therefore, the authors should tone down these aspects in the abstract, in the introduction and in the discussion. Rather, the authors should make a strong statement with their here presented data, that there exists only very week evidence for a physiological beneficial role of the pineal hormone within cardiovascular diseases!

- In the discussion the authors should not put as much emphasis in apologizing for their here presented data, but rather show confidence that their well-designed study reveals the truth!

- the authors should be aware of the fact that the claimed effects of melatonin do not occur when concentration of the pineal hormone stay within the physiological range.

- the authors should be aware of the fact that the halving time of melatonin in the blood circulation is only 20 min. Thus, effects have to occur rapidly or otherwise they are not specific.

- cited studies # 28,29 should be discussed with extreme caution! The authors have done a good job within the design of their study which is in sharp contrast to many other studies!!

MINOR COMMENTS

- Page 1, line 42: erase ‘the pineal derived from’.

- page 2, line 54: misspelling, read ‘Arteriosclerosis’.

Author Response

Title: Melatonin supplementation for 6 weeks had no effect on arterial stiffness and mitochondrial DNA in women aged 55 years and older with insomnia: A double-blind randomized controlled study

Reviewer #2:

The manuscript by Kim and co-workers elaborates on the clinical benefit of supplementation of the pineal hormone melatonin. The authors investigated 38 healthy women, 55 years and older with a recorded history of insomnia. The authors supplemented at nightime only 2mg melatonin, which leads to a serum concentration that is in the upper physiological range. The authors report a beneficial effect on the sleep architecture of participants, and a small effect on blood pressure. However, they could not show any result of melatonin supplementation on arteriosclerotic parameters, nor on mitochondrial DNA copy number. The authors conclude that melatonin has obviously no effect on cardiometabolic parameters. The study is well designed, data are valid and well presented, with minor changes to be incorporated.

  1. In my eyes the anti-oxidative, anti-inflammatory and anti-apoptotic effect of melatonin is largely overestimated. A strong proof of evidence is in my eyes still lacking. Therefore, citations #2-#10 should be taken with caution. I rather recommend that the authors cite solid studies that see the claimed effects of melatonin in a very critical light! While melatonin has a clear-cut effect on the phase of the endogenous circadian clock and on sleep architecture, other attributed effects are standing on week grounds. Therefore, the authors should tone down these aspects in the abstract, in the introduction and in the discussion. Rather, the authors should make a strong statement with their here presented data, that there exists only very week evidence for a physiological beneficial role of the pineal hormone within cardiovascular diseases!

Response: Thank you for your valuable comment. We agree that melatonin’s major effects are to control the endogenous circadian clock and sleep architecture, while its anti-oxidative, anti-inflammatory and anti-apoptotic effects may have weak cardiometabolic effects, which was the main reason we designed this study. Following your suggestions, we toned down strong statements in the revised manuscript as follows.

Abstract:

(1) Melatonin is a hormone produced in the pineal gland that controls sleep and circadian rhythm. Some studies have reported antioxidant and anti-inflammatory effects of melatonin that could benefit cardiometabolic function; however, there is a lack of evidence to support these assertions. The aim of this study was to investigate whether melatonin has beneficial effects on arterial stiffness and mitochondrial deoxyribonucleic acid (DNA) in humans. (e.g. line 16-17 Page 1)

Sleep quality was improved in the melatonin group. Further research, including longer-term studies with higher doses of melatonin, is warranted. (e.g. line 37-38 Page 1)

Introduction:

- The main function of melatonin is to control sleep and circadian rhythm, leading to improved sleep quality.[1] The potential usefulness of melatonin in cardiometabolic dysfunction is a popular topic of research, but in vivo evidence for melatonin benefits beyond sleep improvement remains scarce. Melatonin can play a role in scavenging and oxidative stress.[2] The indoleamine ring of melatonin and its metabolite detoxify free radicals.[3, 4] Biosynthesis of glutathione, a potent antioxidant, is induced by melatonin.[4, 5] Melatonin may also play a role in modulating the immune system and cell aging.[4, 6]

According to a systematic review and meta-analysis by Mohsen Mohammadi-Sartang M, melatonin supplementation significantly reduces triglycerides and total cholesterol levels, which was more evident at higher doses and longer duration.[7] Some studies reported that melatonin supplementation could reduce cardiovascular-metabolic diseases such as hypertension, diabetes mellitus, and obesity.[8-11] Korkmaz A proposed that antioxidant and anti-inflammatory effects, as well as, regulation of the autonomic nervous system (ANS), might play a role in preventing cardiovascular-metabolic diseases.[12] Acuna-Castroviejo D and Leon J suggested that stabilization of mitochondrial function influences arteriosclerosis, a modifiable risk factor for cardiovascular diseases, by regulating chronic low-grade inflammation, oxidative stress and vascular endothelial dysfunction.[13, 14]

We hypothesized that melatonin supplementation could improve cardiometabolic indices in addition to sleep disturbance. Thus, we aimed to investigate the effects of melatonin supplementation on arterial stiffness and cardiovascular risk factors in women aged 55 years and older with insomnia. We also examined quantitative changes in mitochondrial deoxy-ribonucleic acid (DNA) (mtDNA) copy number and autonomic function before and after melatonin supplementation. (e.g. line 1-22 Page 2)

(We added Citation 2 and 7. Previous citation 2 and sentences about previous citation 2 were deleted)

Added citation>

  1. Esposito E, Cuzzocrea S. Antiinflammatory activity of melatonin in central nervous system. Curr Neuropharmacol. 2010 Sep;8(3):228-42.
  2. Mohammadi-Sartang M, Ghorbani M, Mazloom Z. Effects of melatonin supplementation on blood lipid concentrations: A systematic review and meta-analysis of randomized controlled trials. Clin Nutr. 2018 Dec;37(6 Pt A):1943-1954.

Deleted sentences about previous citation 2:

The amount of melatonin secretion may varies with age. The amount is highest at one to three years and gradually decreases with age, which can affect sleep quality in the elderly.[2]

Material and Methods:

After 6 weeks, a physical examination, questionnaire, laboratory tests, and arteriosclerosis and ANS function testing were performed. The study duration was set to six weeks. (e.g. line 43-45 Page 2)

Discussion:

Obayashi[25] and Lee[26] supported the hypothesis that urinary excreted melatonin metabolite levels are inversely associated with arterial stiffness. Initially, we hypothesized that melatonin would improve CAVI as an auxiliary indicator for arterial stiffness [27, 28] and mitochondrial DNA and HOMA-IR as well as the other cardiometabolic markers beyond its effects on insomnia. (e.g. line 29-30 Page 8)

Unfortunately, Arterial stiffness, insulin resistance, and mtDNA copy number were not improved after six weeks of supplementation with melatonin in this study. This is inconsistent with previous studies and our hypothesis. Unlike the abovementioned studies, this study was designed as a randomized, placebo-controlled, double-blinded trial in a hospital setting. It is possible that, compared to observational studies or non-randomized controlled clinical trials, this randomized controlled clinical study is more stringently designed and better reflects the pharmaceutical effects of melatonin in human subjects. Even if melatonin has beneficial effects such as anti-oxidative and anti-inflammatory functions beyond improvement of sleep, pathological conditions such in vivo ROS, in-flammation, and other drugs could overcome melatonin’s benefits.

In part, the origin of these differences may be randomization and allocation at baseline in relation to the small sample size.

(e.g. line 32-41 Page 8)

Our study has some limitations that should be considered when interpreting these results. First, the sample size was too small, and study was conducted at only one large hospital. These make it hard to generalize our findings to the real-world population. Second, Six weeks may be an insufficient duration to observe the effects of improving arterial stiffness, mtDNA, and insulin resistance. Few studies with randomized controlled design have investigate the duration of melatonin use and cardiometabolic function. The follow-up duration of most previous studies on melatonin’s sleep improvement effects was less than four 4 weeks.[11, 20, 29-31] In the few studies that have investigated its cardiometabolic effects, follow-up did not exceed six weeks. Thus, we considered a follow-up duration of six weeks appropriate for investigating the cardiometabolic effects of melatonin. Due to the lack of evidence on long-term use, we set the treatment period to 6 weeks. It is possible that longer-term use could reveal further effects that were not observed in this study, and further research is warranted. Also, a prolonged-release 2-mg melatonin dose might have been too low to reverse arterial stiffness. We could not escalate the dose because of dose restrictions related to Circadin® 2mg and its approval by the Korean Ministry of Food and Drug Safety for melatonin. Future research should investigate higher melatonin doses and their effects on cardiovascular factors and mitochondrial function. In addition, urine metabolites were not measured before and after the experiment to determine the concentration of active melatonin in the human body. Circulating melatonin is inactivated in the liver where it is converted to 6-hydroxymelatonin and then conjugated with sulfate or glucuronide and excreted in urine. Circulating melatonin levels have been reported to decrease with physiologic aging.[32] If the amount of active melatonin was measured, we could more accurately compare the effects on cardiometabolic function of melatonin. Finally, all subjects were generally healthy. Therefore, caution is needed when generalizing these results to people with chronic severe illness.

(e.g. line 45-52, Page 8)

This study has several advantages over previous research. This study was carefully designed as a randomized, placebo-controlled, double-blinded trial that enrolled 38 women. Few randomized, controlled double-blind trials of melatonin supplements have been conducted. Among them, most studies enrolled less than 20 participants. Population size is important because it affects the statistical power. While the number of participants in this study was small, it was larger than previous studies.

(We added this content at line 5, Page 9)

Conclussions

Consistent with previous studies, we have shown that melatonin supplementation for six weeks improves sleep quality. However, it was not proved that melatonin could improve cardiometabolic parameters like arterial stiffness, mtDNA, or insulin resistance in this human study. In part, this null association between melatonin supplementation and cardiometabolic indices might be due to the low dose or short duration of melatonin supplementation. Further investigation with a long-term intervention at a higher dose of melatonin is needed to obtain a clear understanding of the cardiometabolic effects of melatonin treatment.

(e.g. line 11-15 Page 9)

  1. In the discussion the authors should not put as much emphasis in apologizing for their here presented data, but rather show confidence that their well-designed study reveals the truth!

Response: We appreciate your encouraging comments. We carefully described several null associations between melatonin supplementation, cardiovascular factors, and mitochondrial function. As you mentioned, we believe this study has several strengths compared to previous studies. Thus, we would like to insert the strengths in the discussion section as follows.

This study has several advantages over previous research. This study was carefully designed as a randomized, placebo-controlled, double-blinded trial that enrolled 38 women. Few randomized, controlled double-blind trials of melatonin supplements have been conducted. Among them, most studies enrolled less than 20 participants. Population size is important because it affects the statistical power. While the number of participants in this study was small, it was larger than previous studies. (e.g. line 5, Page 9)

  1. The authors should be aware of the fact that the claimed effects of melatonin do not occur when concentration of the pineal hormone stay within the physiological range.

Response: We strongly agree with your comments. According to previous studies, supplementation dose is very important: a low dose of oral melatonin (0.1 to 0.3 mg/d) can reach physiologic melatonin concentration in the body to synchronize the body’s central clock; an intermediate dose (0.6 to 5 mg/d) improves sleep disorders; and a high-dose suppository (300 mg/d) possibly helps neurodegenerative disorders such as amyotrophic lateral sclerosis. Different effects can be expected depending on the administered melatonin dosage. Unfortunately, the Korean Ministry of Food and Drug Safety only officially permitted the 2 mg prolonged release Circadin® for use in sleep disturbance, which is why we could not escalate supplementation dose. This was described in the limitations in the Discussion section. Administration of a higher dose of melatonin (at least 5 mg/d) may exceed the physiologic range and improve cardiovascular factors and mitochondrial function by reaching a concentration in the body where cardiometabolic protection can be expected.

  1. The authors should be aware of the fact that the halving time of melatonin in the blood circulation is only 20 min. Thus, effects have to occur rapidly or otherwise they are not specific.

Response: Thank you for your guidance. The plasma concentration peak from oral melatonin administration arises within 60 minutes. Plasma concentrations diminution is biphasic, with a half-life of 2 and 20 minutes, respectively.

(ref. Tordjman S, Chokron S, Delorme R, et al. Melatonin: Pharmacology, Functions and Therapeutic Benefits. Curr Neuropharmacol. 2017;15(3):434-443.)

However, our melatonin supplement, prolonged-release Circadin®, is similar to the endogenous melatonin release form, and slowly releases melatonin for 8 to 10 hours, working throughout the sleep period without breaking the body's sleep rhythm.

  1. Cited studies # 28,29 should be discussed with extreme caution! The authors have done a good job within the design of their study which is in sharp contrast to many other studies!!

Response: Thank you for your comments. We revised the text to be more cautious about mentioning cardio-ankle vascular index (CAVI) as an atherosclerosis marker in the Discussion section. CAVI is an indicator of arterial stiffness rather than atherosclerosis.

Initially, we hypothesized that melatonin improves CAVI as an atherosclerosis marker [28, 29] (e.g. line 29-30 Page 8)

Thank you for your valuable comment. CAVI is used as a new indicator of arterial stiffness, which does not depend on blood pressure change during measurements. However, as you noted, CAVI is not a standard measurement tool for arterial stiffness.

Thus, we revised the manuscript as follows:

Initially, we hypothesized that melatonin would improve CAVI as an auxiliary indicator for arterial stiffness [28, 29] (e.g. line 29-30 Page 8)

Minor comments:

- Page 1, line 42: erase ‘the pineal derived from’.

- page 2, line 54: misspelling, read ‘Arteriosclerosis’.

=> We revised the script accordingly.